# Validation of a MS Based Proteomics Method for Milk and Egg Quantification in Cookies at the Lowest VITAL Levels: An Alternative to the Use of Precautionary Labeling

**DOI:** 10.3390/foods9101489

**Published:** 2020-10-19

**Authors:** Linda Monaci, Elisabetta De Angelis, Rocco Guagnano, Aristide P. Ganci, Ignazio Garaguso, Alessandro Fiocchi, Rosa Pilolli

**Affiliations:** 1Institute of Sciences of Food Production, CNR-ISPA, 70126 Bari, Italy; elisabetta.deangelis@ispa.cnr.it (E.D.A.); rocco.guagnano@ispa.cnr.it (R.G.); rosa.pilolli@ispa.cnr.it (R.P.); 2PerkinElmer Italia S.p.A., Viale dell’Innovazione 3, 20126 Milano, Italy; aristide.ganci@perkinelmer.com; 3PerkinElmer LAS Germany GmbH, Ferdinand-Porsche-Ring 17, 63110 Rodgau, Germany; ignazio.garaguso@perkinelmer.com; 4Allergy Division, Bambino Gesù Children’s Hospital, Istituti di Ricovero e Cura a Carattere Scientifico, 00165 Rome, Italy; Alessandro.fiocchi@allegriallergia.net

**Keywords:** egg, milk, allergens, multiple reaction monitoring, mass spectrometry, reference doses, food, PAL

## Abstract

The prevalence of food allergy has increased over the last decades and consequently the food labeling policies have improved over the time in different countries to regulate allergen presence in foods. In particular, Reg 1169 in EU mandates the labelling of 14 allergens whenever intentionally added to foods, but the inadvertent contamination by allergens still remains an uncovered topic. In order to warn consumers on the risk of cross-contamination occurring in certain categories of foods, a precautionary allergen labelling system has been put in place by food industries on a voluntary basis. In order to reduce the overuse of precautionary allergen labelling (PAL), reference doses and action limits have been proposed by the Voluntary Incidental Trace Allergen Labelling VITAL project representing a guide in this jeopardizing scenario. Development of sensitive and reliable mass spectrometry methods are therefore of paramount importance in this regard to check the contamination levels in foods. In this paper we describe the development of a time-managed multiple reaction monitoring (MRM) method based on a triple quadrupole platform for milk and egg quantification in processed food. The method was in house validated and allowed to achieve levels of proteins lower than 0.2 mg of total milk and egg proteins, respectively, in cookies, challenging the doses recommended by VITAL. The method was finally applied to cookies labeled as milk and egg-free. This method could represent, in perspective, a promising tool to be implemented along the food chain to detect even tiny amounts of allergens contaminating food commodities.

## 1. Introduction

The most recent epidemiology studies show the continuous increasing prevalence of food allergy worldwide and highlight global disparities of the incidence proportion, influenced by numerous genetic and environmental factors, as well as by gene–environment interactions [1,2]. The main treatment for sensitive individuals appears to be the lifelong avoidance of the offending foods [3]. In order to safeguard the health of sensitive consumers, European Commission Regulation No. 1169/2011 established the list of 14 allergenic ingredients (and by-products) whose presence must be indicated in the respective food labels whenever incorporated into foods. The list includes the following ingredients: milk, egg, cereals containing gluten, fish, crustacean, peanut, soy, tree nuts (hazelnut, almond, walnut, cashew, pecan nuts, Brazil nuts, pistachio, macadamia), sesame, lupin, mustard, celery, mussels, and sulphur dioxide (sulphite) [4]. However, current legislation does not address the unintentional occurrence of allergens due to cross-contamination along the entire food chain, neither established legal threshold levels for managing hidden allergens, posing a relevant health risk to allergic consumers [5]. To fill this gap, various countries have recently set own legal thresholds (e.g., Switzerland, Germany, Belgium, and Netherlands), lacking, however, harmonization among the different legal entities. In this frame, the European project ThRAll, funded by the European Food Safety Agency will actively contribute to the harmonization of MS-based methods by developing a prototype quantitative reference method for the multiple detection of food allergens in incurred food matrices [6].

Since 2007, in absence of official regulatory thresholds and facing the complexity of food allergen management, Australia and New Zealand developed the Voluntary Incidental Trace Allergen Labelling (VITAL) system to assist food producers in managing cross-contamination along the supply chains [7]. This system establishes eliciting doses (EDs) based on clinical studies for the protection of at least 95% (ED05) or 99% (ED01) of allergic people [7,8,9]. Recently, the version 3.0 of the VITAL program was released and for milk and egg proteins it set 0.2 mg total protein of allergenic ingredient as reference dose for action level 1, meaning that below this threshold no precautionary labelling statement is required, and 99% of the allergic population would safely consume the food. To comply with such threshold levels, reliable and sensitive methods are needed for the identification and quantification of allergenic contaminants.

So far, ELISA and PCR represent the techniques most commonly implemented across the laboratories for food allergen control. The limitations affecting these technologies such as cross-reactivity, low inter-assay reproducibility, missing multiplexing ability for ELISA, and the restrictions due to specificity for DNA based method have moved the attention towards LC–MS-based methods, representing a sequence-specific, protein-based approach [10,11,12,13]. Several multiplexing methods using multiple reaction monitoring (MRM) on low resolution mass spectrometers or alternative high-resolution based MS analysis have been reported and recently reviewed [12], all proving the sensitivity and reliability of an MS based analytical approach. Noteworthy, only a few of them were developed and validated on incurred food matrices [14,15,16,17,18,19,20,21,22,23].

The present work aims at evaluating the performance of a targeted multiple reaction monitoring (MRM) MS method using a last generation triple quadrupole mass spectrometer for the simultaneous detection of milk and egg allergens contamination in model bakery products, namely cookies. Synthetic peptides were used for method development and validation. In particular, the cookie reference material (RM) developed by MoniQA Association was used for the estimation of method recovery. This RM was specifically designed to performance evaluation of milk-detection methods and its production mimic as closely as possible the actual manufacturing process. Finally, the developed method was applied to the analysis of real samples to detect milk and egg traces in commercial cookies labelled as “milk and egg allergen free”.

## 2. Materials and Methods

### 2.1. Chemicals and Materials

Solvents and reagents were purchased from Sigma–Aldrich (Milan, Italy) while Trypsin Gold Mass Spectrometry Grade was purchased from Promega (Milan, Italy). Ultrapure water was produced by a Millipore Milli-Q system (Millipore, Bedford, MA, USA) while formic acid (MS grade) was purchased from Fluka (Milan, Italy). Disposable desalting cartridges PD-10 were purchased from GE Healthcare Life Sciences (Milan, Italy) while syringe filters (0.45 µm of porosity in regenerated cellulose RC, and 5 µm of porosity in cellulose acetate CA) were purchased from Sartorius (Gottingem, Germany). Sep-Pak C18 cartridges (50 mg, 1 mL) were obtained from Waters s.p.a. (Milan, Italy). Skim milk powder and whole egg powder were purchased by Sigma Aldrich (Milan, Italy).

For the preparation of matrix matched calibration curves, allergen-free and incurred cookies were produced at laboratory scale according to the recipe already described in a previous paper [19]. The incurred cookie was prepared at a high contamination level and diluted with blank cookie to match the final concentration required.

Cookie reference materials (RM) for milk allergen detection were purchased from MoniQA association (Güssing, Austria). The kit contains the following four samples: (i) a positive control consisting of characterized dried skim milk powder (SMP-MQA 092014) with validated protein content; (ii) a negative control gluten-free cookie (BLANK-MQA 082015), and two incurred materials (gluten free cookies) added with SMP at two concentration levels, (iii) low inclusion level (LOW-MQA 102016, concentration approx. 10 mg_allergenic ingredient_/kg equivalent to 3.54 mg_milk protein_/kg), (iv) high inclusion level (HIGH-MQA 082016, concentration approx. 50 mg_allergenic ingredient_/kg equivalent to 17.7 mg_milk protein_/kg). Ten different lots of blind commercial cookies labeled by the manufacturer as “prepared without adding of milk and eggs” were provided by Galbusera SpA (Cosio Valtellino, Sondrio, Italy).

### 2.2. Synthetic Peptides Standard Solutions

Native synthetic peptides (Appendix A) were synthetized by GenScript (Piscataway, NJ, USA) and distributed by Twin Helix (Milan, Italy). Peptide purity was composed of between 90% and 99% as confirmed by HPLC analysis, while the respective mass was proved by MS analysis. Peptides for each allergen were received as lyophilized powder and reconstituted with 100 mM Bicarbonate Ammonium/Acetonitrile (80/20; *v*/*v*) to reach the concentration of 1 mg/mL. Reconstituted peptides were then aliquoted in a 0.5 mL tube and stored at −20 °C until use.

### 2.3. Sample Preparation Protocol

Firstly, allergen free and incurred cookie prepared at laboratory scale together with commercial cookies were ground mechanically and sifted with a 1-mm sieve. Conditions for total protein extraction, purification, and digestion were described elsewhere [19,20] with few modifications. In particular, the extraction buffer was replaced by Tris–HCl buffer 200 mM with Urea 7 M at pH = 9.2 and the resulting extract was filtered through 5 μm acetate cellulose membranes. Trypsin digestion was stopped after 14 h by acidification (HCl 6 M) and the final digest was centrifuged at 1800× *g* for 10 min before collecting the supernatant. Tryptic digest was then filtered through a 0.45 μm regenerated cellulose (RC) filter and 1 mL aliquot loaded on a C18 SPE column (previously conditioned with methanol and 50 mM ammonium bicarbonate) for a further purification step. C18-retained peptides were washed with 800 µL of 0.1% formic acid aqueous solution and eluted with 1.5 mL of methanol/water (90:10 *v*/*v*). The collected fraction was dried under gentle air stream and suspended in 100 µL of 0.1% formic acid in acetonitrile/water (90/10, *v/v* solution). Samples were finally filtered through a RC 0.45 µm syringe filter. The analytical workflow for sample preparation is schematized in Figure 1.

### 2.4. Liquid Chromatography–Multiple Reaction Monitoring Analysis

LC–MRM analysis was performed on a PerkinElmer UHPLC LX50 System (PerkinElmer Inc., Waltham, MA, USA) coupled with a PerkinElmer QSight^®^ 220 MS/MS (PerkinElmer Inc., Waltham, MA, USA) detector based on triple quadrupole mass analyzer. Peptide mixture (injection volume 10 µL) was separated on a Perkin Elmer Aqueous C18 Column (2.1 × 150 mm; 3 µm; 100 Å) (PerkinElmer Inc., Waltham, MA, USA). LC method parameters are detailed in Appendix A while MRM conditions are summarized in Appendix A. MRM data were acquired in positive ion mode at unit resolution (0.7 ± 0.1 amu) in both Q1 and Q3. ESI source parameters were set as follows: drying gas (nitrogen): 120 (arbitrary units); HSID™ Temp: 250 °C; Nebulizer gas: 300 (arbitrary units); ion source T °C: 400. All instrument control, analysis, and data processing were performed using the Simplicity™ 3Q software platform v. 1.4 (PerkinElmer Inc., Waltham, MA, USA).

### 2.5. Performance Evaluation for In-House Method Validation

#### 2.5.1. Sensitivity

A matrix matched calibration curve was prepared over the concentration range of 0.0125–0.25 µg/mL (four concentration levels) by spiking a defined amount of synthetic peptide stock solutions to tryptic digest of allergen-free cookie extract. All calibration points were filtered using 0.45 μm filters and then injected (10 μL) in duplicate on the column. Native synthetic peptide peak areas were acquired and by applying proper conversion factors (see Figure 2 for details) the reporting units were converted into total proteins of allergenic ingredient (µg/g). Main analytical criteria, such as sensitivity, repeatability/reproducibility, recovery, and processing effect, were evaluated according to these reporting units.

In order to evaluate any eventual effect of processing on the sensitivity of the method, matrix-matched calibration curves prepared by fortifying cookies with allergenic ingredients before processing (incurred samples) were built up for each milk and egg allergen marker selected. Specifically, five concentration levels were prepared in the range 10–300 µg_allergenic ingredient_/g_matrix_. As first level, a cookie incurred at 3000 µg_allergenic ingredient_/g_matrix_ was produced and then submitted to protein extraction and dilution with the blank extract to obtain the point at 300 µg_allergenic ingredient_/g_matrix_. Calibration points at lower concentrations were produced by progressive dilution of the highest level with blank cookies extract. All extracts were then submitted to SEC purification, tryptic digestion, and peptide purification on C18-SPE to be finally filtered on 0.45 μm filters and then injected (10 μL) in duplicate on HPLC/MS equipment. Peptide peak areas were acquired, and the reporting units were converted into total proteins of allergenic ingredient (µg/g) by assuming 35.39% and 48.05% of total protein content for milk and egg ingredients, respectively, in accordance with previous chemical characterization analysis performed on the allergic materials used for cookie production.

#### 2.5.2. Precision

For method precision, a single contamination level at 100 µg_allergenic ingredient_/g_matrix_ was analyzed. Five analytical replicates were prepared and analyzed (intra-day repeatability). The same analyses were repeated over three different days and compared by one-way ANOVA test at 95% confidence level.

#### 2.5.3. Trueness

Method recovery was evaluated only for milk by means of the validated RMs developed by MoniQA association. The blank sample provided with the kit was used to create a new matrix-matched calibration curve with synthetic peptides. The LOW and HIGH incurred samples were analyzed and the percent ratio between the measured and the validated concentration values defined the method recovery.

## 3. Results and Discussion

### 3.1. Optimization of LC–MS Instrumental Conditions

A sensitive method based on HPLC separation and mass spectrometry detection equipped with triple quadrupole analyzer for the simultaneous detection of milk and egg allergens in a model bakery product, namely cookie, was developed. The proteomic bottom-up approach was applied by detecting proteotypic peptides for monitoring food contamination by allergenic ingredients. Both milk and egg are widely investigated allergens and as such, a good consensus about the most reliable peptide markers has been achieved already by independent investigations [24]. The peptides that arose from tryptic digestion of αS1-casein, namely FFVAPFPEVFGK (FFV) and YLGYLEQLLR (YLG), and from β-lactoglobulin, namely TPEVDDEALEK (TPE) and VLVLDTDYK (VLV) were used for tracking milk, and peptides belonging to ovalbumin, ISQAVHAAHAEINEAGR (ISQ) and GGLEPINFQTAADQAR (GGL) and to vitellogenin-II namely NIPFAEYPTYK (NIP) and NIGELGVEK (NIG) were chosen for egg detection. All these markers have been already validated by previous works accomplished on bakery products [24,25,26].

In order to build up an analytical method for absolute quantitation, synthetic analogous of the aforementioned peptide sequences were purchased. Firstly, standard solutions of such peptides were prepared and injected in flow analysis for the optimization of instrumental parameters setting up the MRM detection on triple quadrupoles. For each peptide, the three most sensitive transitions were selected and collision energies, entrance voltages, and collision cell lens voltages were tuned to maximize the signal to noise ratio (Appendix A). The chromatographic conditions for peptide separation were optimized and the best compromise between total running time and peak resolution was found.

In order to confirm the absence of interfering peaks from the matrix background, a blank cookie sample was prepared according to the sample preparation protocol described in Section 2.4 and added with synthetic peptides at fixed concentration. In Figure 3, a typical chromatogram acquired under the best separation conditions is presented and averaged peak retention times are reported.

### 3.2. Sensitivity and Matrix Effect

After optimizing the instrumental conditions, different aliquots of blank cookie samples were added with increasing concentration of milk and egg synthetic peptides in order to build-up matrix-matched calibration curves. In particular, four calibration points within the range of 0.125–0.25 µg/mL were prepared and the linear interpolation of resulting peak areas allowed evaluating the linearity range, and the sensitivity for each precursor/transition acquired.

One of the controversial aspects in food allergen detection has been the reporting unit of the contamination level. As well known the legislation refers to allergen labelling as whole ingredient, although clinical studies and potential threshold levels refer to the total protein content of the allergenic ingredient.

Specifically, protein is the hazard that causes allergic reactions, therefore analytical methods reporting contamination level as mg of total proteins would streamline the usability of the information retrieved also in light of the adherence to prescribed threshold levels and of the consistency of method sensitivity to reference doses of the VITAL Program. This issue represents an important bottleneck for mass spectrometric detection where for absolute quantitation, peptide-based calibration curves, and further conversions from the peptide units into total protein units, are required. Practically, in order to calculate final protein concentration, the peptide concentration in the digest volume (µg/mL) needs to be converted into total protein of the allergenic ingredient in matrix weight (µg/g). Until now, no international agreement about proper conversion factors has been achieved and only few examples from previous literature are to date available [15,23] on this regard. In this investigation, we applied a similar conversion scheme presented in Figure 2. Both milk and egg have been widely investigated in terms of protein composition, therefore the information available in the literature was used to retrieve proper conversion factors based on specific mathematical calculation and molar equivalence as schematized in Figure 2. Briefly, for each synthetic peptide, peptide concentration in the tryptic digest (reported as µg/mL) was first converted into molarity and then, assuming the complete release of each peptide from its parent protein, protein molarities were calculated. Afterwards, based on protein molecular weight and its relative abundance within the total proteins contained in food ingredient we calculated the total allergen proteins per mL of digest. Finally, as last conversion step, by taking into consideration the solid/liquid ratio used for sample extraction (1:20), we obtained the required reporting unit of µg of total protein of the allergenic ingredient per g of matrix. By following this approach all peptide reporting units were converted into µg_total protein_/g_matrix_ providing an analytical range between 1.3 and 680 µg/g depending of the specific marker. The new reporting units were integrated in the matrix-matched calibration curves and all method performance features were referred to them. The response linearity obtained in the matrix-matched calibration curve was very good for all the peptide markers monitored within the investigated range, with linear correlation coefficients at least better than 0.9859. Limit of detection (LOD) and quantification (LOQ) were calculated according to the interpolation parameters as 3-times and 10-times, respectively, the standard deviation of the line intercept divided by the slope. The careful evaluation of LOD/LOQ values for the detected transitions allowed to identify the best quantifier marker and its most sensitive transition as reported in Table 1.

These analyses allowed us to further evaluate the matrix effect on the peptides chosen. According to our results, very challenging LODs were achieved, as low as 0.1 and 3 µg_tot prot_/g_matrix_, respectively, and referred to FFV and TPE αS1-casein and β-lactoglobulin peptides for milk allergen. As for egg, LODs of 0.3 and 3 µg_tot prot_/g_matrix_ were found for ISQ (ovalbumin) and NIP (vitellogenin-2) peptides, respectively. LOQ values are also reported in Table 1. Noteworthy, the sensitivity provided by the peptides TPE and NIP was lower than the peptides FFV and ISQ, respectively; however, it is important to keep them in the analytical method as specific markers of whey and yolk proteins, notwithstanding their lower relevance from the allergological point of view, to encompass also risk of contamination from partial milk/egg based formulations. In Figure 4, it is shown a typical chromatogram obtained for the milk (FFV-*m*/*z* 692.9→991.4 and TPE-*m*/*z* 623.3→572.5) and egg allergens (ISQ-*m*/*z* 592.1→858.9 and NIP-*m*/*z* 671.8→557.9) and their relevant confirmative transitions in the cookie sample.

### 3.3. Sensitivity of the Method in Incurred Cookies and Compliance with the VITAL Reference Doses

As already mentioned, the VITAL grid was developed in 2007, aiming at providing a helpful management tool for food producers as well as to consumers. Although originally created by the Allergen Bureau of Australia and New Zealand, this system has been taken into consideration and used as reference values by numerous countries within the European Union until other official and harmonized limits will be available for the different allergenic foods. In particular, the VITAL Program provides a quantitative method for risk-assessment to evaluate the impact of allergen cross-contamination and to make decisions regarding proper precautionary allergen management and labeling. This approach allows not only safeguarding the health of allergic consumers, but also preserving the value of precautionary labeling as a risk management tool, avoiding its massive use also in very low-risk cases. The likelihood to develop an adverse reaction in allergic people depends on the total amount of allergenic proteins consumed during a meal, and on the level of sensitization of each individual. Therefore, the crucial point in the VITAL Program was to find a correlation between these two topics and define the maximum concentration level from accidental contamination that does not present a risk for most of the allergic population (95% or 99%, depending on data) according to clinical data available of minimum eliciting doses. Above these reference doses, precautionary labelling warning of potential cross-contamination is required.

VITAL system relies on three key values. First, the “reference amount” that represents the portion size, namely the maximum amount of a food eaten in a typical eating occasion. Second, the “reference dose” which refers to the protein level (total protein in milligrams from an allergenic food) below which only the most sensitive individuals (between 1 and 5) in the allergic population are likely to experience an adverse reaction. Third, the “action levels” that are threshold levels of protein concentrations in food guiding the labelling (action level 1: no precautionary labelling required, action level 2: “may contain” labelling required, and action level 3: “contain” labelling required).

The latter are calculated according to the set reference dose and reference amount, becoming part of action level grids for easy use by food producers.

Starting from this, in order to provide useful tools for food allergen risk-management, good sensitivity is demanded for new analytical methods complying with the action levels prescribed by the VITAL program. Such levels are periodically updated according to new allergological data available from clinical studies, and last values of VITAL program version 3.0, were revised and released in October 2019. As for milk and egg, an equal reference dose of 0.2 mg_total protein_ was set, and referred to a portion size of 50 g, deemed reasonable for cookies, thus resulting in an action level 1 of 4 mg_total protein_/kg.

Noteworthy, by applying this method following two different routes we obtained two different sensitivities according to the type of allergen contamination occurring in the food matrix. As for incurred cookies the method reached sensitivity down to 4 mg_total protein_/kg this one being the minimum level detectable by the method in use and in compliance with the VITAL sensitivities required. This limit might then represent the highest protection level offered to the allergic patient since it refers to a cookie incurred at the beginning of the whole process taking into account the processing effect as well as the extraction efficiency of the containing proteins.

### 3.4. Precision

Intra-day and inter-day precision of the analytical method (percent coefficient of variation in peak areas at a fixed concentration, CV%) were evaluated to test the method repeatability and reproducibility within the same laboratory. To this purpose, a blank cookie sample fortified with skim milk and whole egg powders at the final level of 100 µg_allergenic ingredient_/g_matrix_ was prepared. The intra-day repeatability was calculated within five independent replicates and values lower than 10% were obtained in all cases, with the best repeatability provided by the αS1-casein marker FFV and the ovalbumin marker ISQ, due to the high abundance of these proteins in the allergic ingredients. On the contrary, inter-day repeatability was calculated over 3 days by analyzing the same fortified samples. Obtained values were always lower than 9% for both milk and egg quantifier peptides. The mean values obtained on different days were compared by a one-way ANOVA test at 95% confidence level, resulting in no significant differences for all peptide markers.

### 3.5. Evaluation of Processing Effect on Method Sensitivity

As known, food processing can deeply affect the structure and stability of a protein as well as its solubility due to several chemical modifications that can occur during thermal treatment. Consequently, the analytical detection can be affected as well when extensively processed foods are investigated for allergen contamination. In order to evaluate the effects of food processing on the detection of each milk and egg peptide marker, specific matrix matched calibration curves were obtained by progressive dilution of incurred cookies extract fortified with milk and egg allergens at high level. As known, incurred material is produced by adding allergic ingredients during dough preparation and before thermal treatment. This condition reproduces what is actually happening during food processing leading to a more reliable estimation of method sensitivity considering the overall effects of processing on protein stability and solubility. As a result, the final recovery and performance of the method could be taken into account. As detailed in Section 2, incurred-cookie calibration curves were produced within a certain concentration range. Following, on the basis of the total protein contents estimated for skim milk powder and whole egg materials used for the preparation of incurred cookies (35.39% and 48.05% of total protein content for milk and egg, respectively) all the peptide reporting units were converted into µg_total protein_/g_matrix_ providing an analytical range between 3.5 and 106.2 µg/g for milk proteins and 4.8 and 144.2 µg/g for egg. By linear interpolation of the calculated peak areas we retrieved information on the linearity range and the sensitivity for each precursor/transition acquired. Results are depicted in Table 2. LODs of 1.6 and 3.5 µg_total protein_/g_matrix_ were calculated for FFV and TPE quantifier milk peptides while higher LODs were obtained for egg allergen, namely 4.0 and 4.8 µg_total protein_/g_matrix_ for ISQ and NIP quantifier egg peptides. By comparing LODs calculated for synthetic peptide-curve calibrations with incurred cookie-curve calibrations, a sensitivity reduction of approximately 94% and 97% was observed both for FFV and ISQ milk and egg peptides due to processing effect, while a slight reduction (of 14% and 38%, respectively) was calculated for TPE (whey proteins) and NIP (yolk proteins) peptides. These results are in accordance with our previous investigation [20] where a reduction of milk and egg detection sensitivity of approximately 93% and 97% were recorded for milk (based on casein marker) and egg allergens (based on white egg marker). The data gathered can be explained by taking into account the labile behavior shown by specific proteins during some processing applied to food [20].

### 3.6. Trueness

Trueness of the method was evaluated by performing dedicated experiments on the only reference material available on the market validated for milk detection in cookie matrix. The purchased kit contains two samples at different concentration levels, namely 10 and 50 µg_allergenic ingredient_/g_matrix_, that correspond according to the certificate of analysis, to 3.54 and 17.7 µg_tot prot_/g_matrix_, respectively. Trueness evaluation was limited to milk allergen since no reference materials for baked food are available yet for egg allergen. Specifically, incurred reference materials were subjected to the whole sample preparation along with reference allergen-free cookie sample. The latter was used to build-up new matrix-matched calibration curves with synthetic peptides covering the range 1–50 µg_tot prot_/g_matrix_, and calculate line equation in the specific cookie matrix provided with the kit. The low and high incurred samples were both analyzed in triplicate (independent samples), and experimental concentration values obtained by curve interpolation were compared with theoretical ones. The percentage ratio between the experimental and theoretical values provided an estimate of the method recovery for milk allergen. Method recovery calculated with YLG peptide was 57 ± 6%, and 56 ± 7% at low and high concentration levels, respectively, whereas the recovery calculated with the peptide FFV was 57 ± 4% and 50 ± 3% at low and high concentration levels, respectively (see Appendix A).

### 3.7. Occurrence of Milk and Egg Contamination in Commercial Samples Declared “Prepared without Adding Milk and Egg”

In the final part of the work, the validated method was applied to samples taken from 10 different lots of commercial cookies and labelled as “prepared without adding of milk and eggs” in order to assess the actual absence of any trace of milk and egg allergens, according to the sensitivity of the method.

Cookies were submitted to sample preparation and analyzed in duplicates with the analytical method herein described and optimized. No quantifiable peaks areas were detected for milk and egg quantifier peptides, therefore, we concluded that no accidental contamination occurred in these samples, at least within the sensitivity limits reported by the developed method. The analytical method in-house validated in the present work, demonstrated to be a sensitive tool for the quantification of egg and milk allergens in cookies at the highest confidence level in compliance with the VITAL doses recommended. This approach, in perspective, can represent a valid alternative to the use of PAL.

## 4. Conclusions

The method herein described based on QSight triple quadrupole mass analyzer provides an optimized sample preparation protocol and a MRM method for the simultaneous quantification of egg and milk in cookies selected as a model bakery product. Method performance was assessed by using selected milk and egg synthetic peptide markers and a proper factor to convert peptide into protein concentration was proposed. The LOD and LOQ values obtained for both egg and milk allergens calculated in incurred cookies (referred to the protein content) allowed to detect levels of contamination complying with the reference thresholds set for egg and milk and recommended (action level 1) by the VITAL program v 3.0. Additionally, method precision provided good results for both the allergenic ingredients analyzed in this matrix. Processing effects were also assessed confirming previous evidence about the reduced detectability for both allergens, with milk proteins being more susceptible to thermal processing effect. To the best of our knowledge, this is the first time the trueness of the method was calculated by means of MoniQA reference material in this type of food material. The in-house validation performed provided analytical features that complied with the minimum requirements set in the AOAC SMPR 2016.002 for allergen detection in food. Finally, the method was also challenged with real samples from the market to test its realistic potential in detecting accidental cross-contamination in real samples. Commercial cookies labeled as “milk and egg ingredients free” were analyzed by exploiting the developed method and none of them were found incorrectly labeled, within the sensitivity limits achieved with this method. In perspective, the multi-allergen MS based method developed can be employed for allergen control in food supply chains where cross-contamination is likely to occur, hence avoiding the resort to PAL.

## Figures and Tables

**Figure 1 foods-09-01489-f001:**
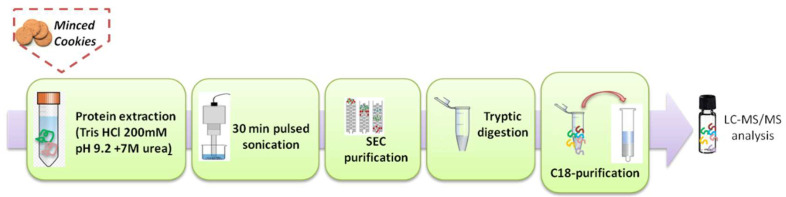
Experimental workflow for the simultaneous detection of milk and egg allergens in cookie samples.

**Figure 2 foods-09-01489-f002:**
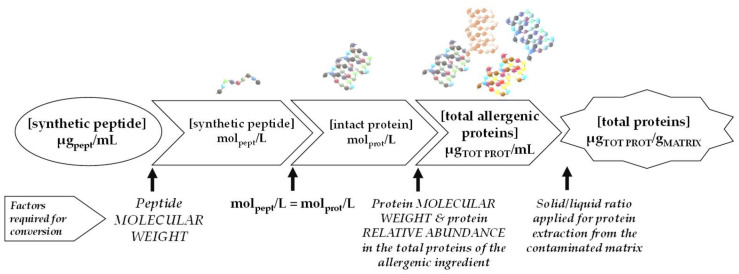
Flowchart calculation for the conversion of egg and milk synthetic peptides concentration (µg_peptide_/mL_extract_) into total protein concentration (µg_tot prot_/g_matrix_).

**Figure 3 foods-09-01489-f003:**
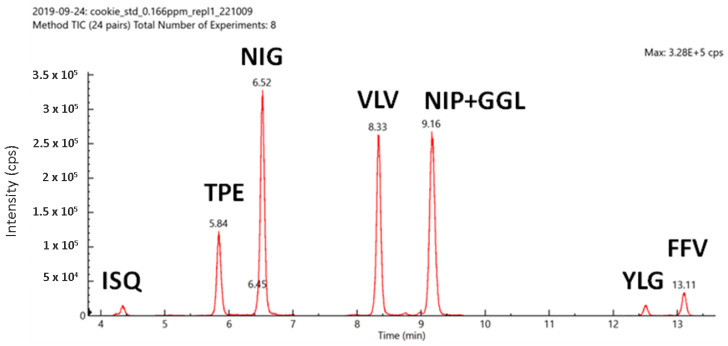
Typical chromatograms recorded for synthetic peptides in cookie matrix (total ion current, peptide concentration level 0.166 µg/mL).

**Figure 4 foods-09-01489-f004:**
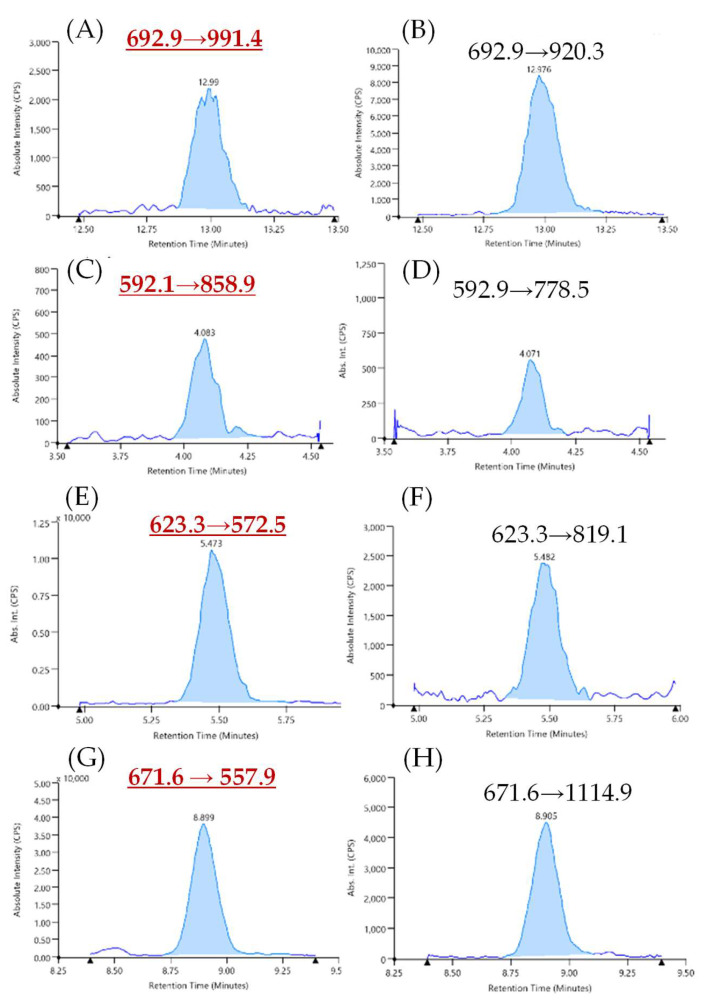
Typical chromatograms acquired for synthetic peptides in the cookie matrix: Extracted Ion Chromatogram XIC of quantifier transitions for most sensitive marker peptides of milk ((**A**), FFV = *m*/*z* 692.9→991.4; (**C**), TPE = *m*/*z* 623.3→572.5) along with their relevant qualifier transition ((**B**), FFV = *m*/*z* 692.9→920.3; (**D**), TPE = *m*/*z* 623.3→819.1) and egg, quantifier transitions ((**E**), ISQ = *m*/*z* 592.1→858.9; (**G**), NIP = *m*/*z* 671.6→557.9); qualifier transitions ((**F**), ISQ = *m*/*z* 592.1→778.5; (**H**), NIP = *m*/*z* 671.6→1114.9) at a level of 0.0125 µg/mL.

**Table 1 foods-09-01489-t001:** Analytical features of the developed analytical method on the basis of synthetic peptides matrix-matched calibration curves.

Allergenic Ingredient: Protein	Marker	Quantifier Transition	LOD/LOQ (µg_TOT PROT_/g_MATRIX_)	*R* ^2^	CV% _intra_ Day 1	CV% _intra_ Day 2	CV% _intra_ Day 4	CV% _inter_	Recovery LOW-MQA Material	Recovery HIGH-MQA Material
Milk: αS1-Casein	FFV	692.9→991.4	0.10/0.3	1.0000	1.9%	0.7%	6%	4%	57 ± 4%	50 ± 3%
Milk: β-Lactoglobulin	TPE	623.3→572.5	3/8	0.9992	8%	9%	10%	9%	-	-
Egg: Ovalbumin	ISQ	592.1→858.9	0.3/1.1	1.0000	2%	5%	1.7%	4%	-	-
Egg: Vitellogenin-2	NIP	671.8→557.9	3/9	1.0000	3%	3%	4%	6%	-	-

**Table 2 foods-09-01489-t002:** Relevant parameters of milk and egg peptides referred to matrix-matched calibration curves built up by using incurred cookies.

Allergenic Ingredient: Protein	Marker	Quantifier Transition	LOD/LOQ (μg_TOT PROT_/g_MATRIX_) Incurred Material	*R* ^2^
Milk: α-S1-casein	FFV	692.9→991.4	1.6/5.4	0.9969
Milk: β-Lactoglobulin	TPE	623.3→572.5	3.5/11.7	0.9854
Egg: ovalbumin	ISQ	592.1→858.9	4/15.6	0.9903
Egg: vitellogenin	NIP	671.8→557.9	4.8/14.0	0.9896

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
