# Peer review of "Validation of a MS Based Proteomics Method for Milk and Egg Quantification in Cookies at the Lowest VITAL Levels: An Alternative to the Use of Precautionary Labeling"

_foods, 2020, doi:10.3390/foods9101489_

Round 1
Reviewer 1 Report
General comments:
The manuscript by Linda Monaci et al. number ID: foods-929225 “Validation of a MS based proteomics method for milk and egg quantification in cookies at the lowest VITAL levels: an alternative to the use of precautionary labeling”, present for the first time the development of a MRM method on a triple quadrupole instrument for the milk and egg quantification in processed food as cookies. The method allows the detection of proteins lower than 0.2 mg of total milk and egg as doses recommended by the Voluntary Incidental Trace Allergen Labelling (VITAL) program necessary to assist to food producers in order to avoid cross-contaminations. I do not have any major concerns about the study. The manuscript is well organized and written, once the authors have dealt with some minor points detailed below.
Specific comments:
1.- Line 133 please indicate the amount of peptides that were in injected.
2.- Please improve the quality of Figures 2 and 4.
3.- Please indicate the transitions in the manuscript and in the figures as:
parent m/z → fragment m/z
Author Response
Reviewer # 1
General comments:
The manuscript by Linda Monaci et al. number ID: foods-929225 “Validation of a MS based proteomics method for milk and egg quantification in cookies at the lowest VITAL levels: an alternative to the use of precautionary labeling”, present for the first time the development of a MRM method on a triple quadrupole instrument for the milk and egg quantification in processed food as cookies. The method allows the detection of proteins lower than 0.2 mg of total milk and egg as doses recommended by the Voluntary Incidental Trace Allergen Labelling (VITAL) program necessary to assist to food producers in order to avoid cross-contaminations. I do not have any major concerns about the study. The manuscript is well organized and written, once the authors have dealt with some minor points detailed below.
Specific comments:
1.- Line 133 please indicate the amount of peptides that were in injected.
2.- Please improve the quality of Figures 2 and 4.
3.- Please indicate the transitions in the manuscript and in the figures as:
parent m/z → fragment m/z
We thank the Reviewer for the positive comments on the Manuscript. All the minor requests were taken into consideration for revision. As for Figure 4 we improved the resolution as much as we can, but please consider that the original integrated peak can be exported only in this format by the Quantification software.
Reviewer 2 Report
Authors have designed and validates a proteomic method for the detection and quantification of milk and eggs allergens, based on the data recommended by VITAL 3.0 Program. As stated by the authors, it would be an alternative of Precautionary Allergen Labelling as well as other common detection methodologies based on proteins or nucleic acids.
This reviewer finds the paper very interesting for the scientific community, because of its content and methodology. It is well designed and discussed, and perfectly described. It is clear and detailed, and methodology mistakes are not detected. Figures and tables provide enough information about the results, and figure quality seems appropriate. I recommend its publication in Foods Journal in its present form, after minor spell checking.
Author Response
Authors have designed and validates a proteomic method for the detection and quantification of milk and eggs allergens, based on the data recommended by VITAL 3.0 Program. As stated by the authors, it would be an alternative of Precautionary Allergen Labelling as well as other common detection methodologies based on proteins or nucleic acids.
This reviewer finds the paper very interesting for the scientific community, because of its content and methodology. It is well designed and discussed, and perfectly described. It is clear and detailed, and methodology mistakes are not detected. Figures and tables provide enough information about the results, and figure quality seems appropriate. I recommend its publication in Foods Journal in its present form, after minor spell checking.
We thank the Reviewer for the positive comments on the Manuscript.
Reviewer 3 Report
I have no real comments. The MS is written to a high standard and the flow and content of the presentation is balanced and is very good.
A very minor point I found one spelling error of 'traking' instead of 'tracking'. Another minor point is whether percentages to 4 significant figures have that accuracy.
Author Response
We thank the reviewer for being positive and rupporting abou the publication.
The few minor details were amended as suggested.
Reviewer 4 Report
This study aimed to develop a validation of a MS based proteomics method for milk and egg quantification in cookies. However, it is a technical method, not an academic research. I will have no better suggestions.
Author Response
We thank the reviewer for his comment. Indeed this is a technical paper but with important implication on the food control and food safety assessment.
We have produced a slightly revised version according to the comments received.